

# Bivalve microbiomes are shaped by host species, size, parasite infection, and environment

Sarah Gignoux-Wolfsohn[1,2], Monserrat Garcia Ruiz[2],
Diana Portugal Barron[2,3], Gregory Ruiz[4] and Katrina Lohan[2]

[1] Biological Sciences, University of Massachusetts at Lowell, Lowell, MA, United States
[2] Coastal Disease Ecology Laboratory, Smithsonian Environmental Research Center, Edgewater, MD, United States
[3] Department of Neurology, Brain Research Institute, Mary S. Easton Center for Alzheimer's Research and Care, David Geffen School of Medicine, University of California, Los Angeles, Los Angeles, CA, United States
[4] Marine Invasions Laboratory, Smithsonian Environmental Research Center, Edgewater, MD, United States

Corresponding author
Sarah Gignoux-Wolfsohn,
sarah_gignouxwolfsohn@uml.edu

## ABSTRACT

Many factors affect an organism's microbiome including its environment, proximity to other organisms, and physiological condition. As filter feeders, bivalves have highly plastic microbiomes that are especially influenced by the surrounding seawater, yet they also maintain a unique core set of microbes. Using 16S ribosomal RNA sequencing, we characterized the bacterial microbiomes of four species of bivalves native to the Mid-Atlantic East Coast of North America: *Crassostrea virginica, Macoma balthica, Ameritella mitchelli*, and *Ischadium recurvum* and assessed the impact of their external environment, internal parasites, and size on their microbial communities. We found significant differences in bacterial amplicon sequence variants (ASVs) across species, with each species harboring a core ASV present across all individuals. We further found that some *C. virginica* co-cultured with *I. recurvum* had high abundances of the *I. recurvum* core ASV. We identified ASVs associated with infection by the parasites *Perkinsus marinus* and *Zaops ostreum* as well others associated with bivalve size. Several of these ASV are candidates for further investigation as potential probiotics, as they were found positively correlated with bivalve size and health. This research represents the first description of the microbiomes of *A. mitchelli, I. recurvum*, and *M. balthica*. We document that all four species have highly plastic microbiomes, while maintaining certain core bacteria, with important implications for growth, health, and adaptation to new environments.

## INTRODUCTION

Host-associated microbes play critical roles in host fitness and ecosystem function (*Gilbert, Sapp & Tauber, 2012*). Microbes can provide hosts with necessary compounds (*Dubilier, Bergin & Lott, 2008*), protect against pathogens (*Teixeira, Ferreira & Ashburner, 2008*),

and metabolize nutrients (*Oliphant & Allen-Vercoe, 2019*). While some symbiotic relationships between animals and bacteria are highly conserved and well-understood (*e.g.*, *Wolbachia*'s transfer of pathogen resistance to various arthropod species (*Kaur et al., 2021*), or the *Vibrio* symbiont that affects bobtail squid organ development (*Nyholm & McFall-Ngai, 2004*), animal microbiomes contain untold numbers of bacterial strains whose effects on host fitness remain unknown. Hosts can acquire microbes both vertically (from parents) and horizontally (from their surrounding environment; *Bright & Bulgheresi, 2010*). Individuals of the same species will therefore share certain species/strains (core microbes), while other strains vary with environmental parameters (*Spor, Koren & Ley, 2011*), such as surrounding location (*Woodhams et al., 2020*), season (*Guo et al., 2021*), and even time of day (*Becker et al., 2020*). This high variability challenges our ability to understand the nature of host-symbiont relationships including how individual microbes affect and are affected by host health. Efforts to describe patterns of microbe distribution and organization across individuals and locations provide a needed first step to characterize host-microbe relationships.

Bivalves are an especially useful system in which to explore these topics as they are filter feeders, which acquire microbes from the surrounding seawater and also maintain unique and distinct bacterial microbiomes (*Arfken et al., 2021*). Furthermore, many bivalves are sessile, live in close proximity to other closely related species, and are therefore highly impacted by their immediate environment and community. Bivalve microbiomes can be influenced by the environment, with multiple studies demonstrating that microbiomes vary across geographic locations (*e.g.*, *King et al., 2012*, *2019*; *Lokmer et al., 2016*; *Stevick, Post & Gómez-Chiarri, 2021*) as well as with differences in environmental factors such as temperature (*Lokmer & Mathias Wegner, 2015*), herbicides (*Britt et al., 2020*), and dissolved oxygen (*Khan et al., 2018*). In addition, bivalve microbiomes can be altered by disease (*King et al., 2019*) and infection by microbial pathogens (*de Lorgeril et al., 2018*) and other parasites (*Green & Barnes, 2010*). However, bacteria present in the microbiome can also protect against pathogens and increase bivalve fitness therefore enabling the development of probiotics as a sustainable treatment for farmed bivalves (*Kesarcodi-Watson et al., 2012*; *Yeh et al., 2020*; *Sumon et al., 2022*). Understanding the factors that shape bivalve microbiomes is important to understand not just the health of the animal itself, but the broader ecosystem function and ecosystem services provided by these animals.

Here, we characterize the microbiomes of four Mid-Atlantic bivalves (*Crassostrea virginica, Macoma balthica, Ameritella mitchelli,* and *Ischadium recurvum*) to explore the impact of biotic and abiotic factors on bacterial diversity and composition. The eastern oyster, *C. virginica* is the dominant bivalve on the east coast of North America and the most well-studied of our host species. In the northern parts of its range, *C. virginica* builds reefs that provide habitat for numerous marine animals and protection for human coastal communities (*Grabowski et al., 2012*; *Hoellein & Zarnoch, 2014*). *Crassostrea virginica* is also economically important as both a wild-harvested and farmed species (*Lipton et al., 2019*; *Parker & Bricker, 2020*). These oysters and their microbiomes play a crucial role in the ecosystem as denitrifiers (*Ray & Fulweiler, 2020*; *Arfken et al., 2017*), with live

symbiont-containing oysters denitrifying more than sediment or oyster shell alone (*Smyth et al., 2016*).

Throughout their range, oysters live adjacent to other bivalves, including other oysters, clams, and mussels (*Coen & Luckenbach, 2000*). These other bivalves can serve important roles from both an ecological (*e.g.*, as prey (*Zwarts & Blomert, 1992*)) and ecosystem services perspective (*e.g.*, denitrification (*Smyth et al., 2018*), and water quality restoration (*Gedan, Kellogg & Breitburg, 2014*)). The hooked mussel, *I. recurvum*, lives sympatrically attached to oysters' shells in low salinity areas in the mid-Atlantic. The two clams *A. mitchelli* and *M. balthica* are thin-shelled clams that live buried in soft sediments often adjacent to oyster reefs north of Cape Hatteras on the East Coast of the U.S. They are crucial prey for multiple estuarine species including the economically and ecologically important blue crab *Callinectes sapidus* (*Laughlin, 1982*; *Hines, Haddon & Wiechert, 1990*; *Seitz, Lipcius & Seebo, 2005*). We lack a basic understanding of the interplay between these bivalves and reefs of *C. virginica*, including how related their microbiomes are, limiting our understanding of their role in maintaining estuarine health. The novel classification of the microbiome of these two clam species will expand our understanding of their basic biology and how it is influenced by their environment.

Mid-Atlantic bivalves are affected by multiple protozoan and metazoan parasites, some of which cause disease and mass mortalities (*Aguirre-Macedo et al., 2007*; *Arzul & Carnegie, 2015*). One of the most widespread groups of protozoan parasites is the genus *Perkinsus* (*Dungan & Reece, 2020*). *Perkinsus marinus* (Dermo) infects *C. virginica* across almost its entire geographic range (*Lohan et al., 2016*, *2018*; *Fernández Robledo et al., 2018*). While *P. marinus* epizootics can and have resulted in the collapse of entire ecosystems (*Hewatt & Andrews, 1954*; *Mackin, 1951*; *Ray, 1954*), enzootic populations are currently common, with mortality from infections occurring after many years (*Calvo et al., 2003*). Relationships between *C. virginica* microbiomes and infection by *P. marinus* have been found previously (*Sakowski, 2015*; *Pimentel et al., 2021*).

Pea crabs like *Zaops ostreum* are small parasitic crabs that live inside bivalve shells and eat plankton concentrated by the host (*Orton, 1920*). While *Z. ostreum* do not cause bivalve mortality, the presence of these pea crabs significantly decreases the condition index of oysters (*Hanke et al., 2015*; *Watts et al., 2018*). Relationships between *Z. ostreum* and bacteria remain uncharacterized.

We conducted 16S sequencing on gill, mantle, and digestive gland from four bivalve species: *C. virginica*, *I. recurvum*, *M. balthica* and *A. mitchelli*. In addition, using organisms that were part of two large-scale manipulative experiments, we examined the effects of field location, parasite (*P. marinus*, *Z. ostreum*) infection, and bivalve community on the *C. virginica* microbiome. Our goal was to elucidate possible connections between bivalve microbial communities and both intrinsic (species, growth) and extrinsic (location, parasites, and bivalve community) factors for these four species of bivalves. Understanding the factors that shape bivalve microbiomes is important to understand not just the health of the animal itself, but the broader ecosystem function and ecosystem services provided by reefs.

## MATERIALS AND METHODS

### 2017 experiment

To examine the effects of parasite infection and location on the oyster microbiome, an experiment was conducted in May 2017 where naive 1 year old diploid seed oysters purchased from Cherrystone Aquafarms (Cape Charles, Virginia, USA) were outplanted onto local natural existing oyster reefs at three sites (North of Wachapreague, South of Wachapreague, and Oyster, VA; Fig. S1) with known high prevalence of *Perkinsus marinus* and *Zaops ostreum* (*Burreson & Calvo, 1996*). Oysters were placed in 16 mesh bags at each site (10 oysters per bag) secured to the reef by a cinderblock and rebar (Fig. S2). Prior to deployment, individual oysters were weighed (total wet weight) and measured (maximum length from valve to edge) and placed in a marked large mesh 63 cm$^2$ container within the oyster bag for future identification. Salinity ranged between 27 and 32 ppt. Note: while the initial intention of the experiment had been to manipulate density and cohabitation as in the 2018 experiment described below, none of the treatments were successfully applied and references to low/high and poly/mono in the unique id of oysters are therefore meaningless and serve only as a unique code.

Oysters were removed from the reefs in September 2017 and again weighed and measured. All surviving oysters in each bag were then shucked using a sterile oyster knife, and tissue (mantle, gill, and digestive gland, avoiding the stomach) was placed in a cryovial with 95% ethanol for future microbiome analysis. Presence of *Z. ostreum* was noted. To determine infection with *Perkinsus spp.*, a piece of rectum and mantle tissue was placed in Ray's Fluid Thioglycollate Media (RFTM; *Ray, 1966*). We incubated the tissue in RFTM at 27 °C for 5–7 days. Incubation in RFTM causes viable *Perkinsus* spp. cells to enlarge making them visible under a dissecting microscope. We placed tissue on a microscope slide and stained with 2–5 drops of Lugol's iodine (Sigma Aldrich, St. Louis, MO, USA), allowing visualization of cells. We quantified intensity of infection with the Mackin Scale, which quantifies infection intensity on a scale from 1 to 5, with 0 indicating no infection (*Ray, 1954*).

### 2018 experiment

To examine the effects of bivalve cohabitation on microbiome composition, we conducted a manipulative mesocosm experiment in the summer of 2018 at the Smithsonian Environmental Research Center (SERC) in Edgewater, MD. Flow-through raw seawater from the Rhode River was continuously fed to five-gallon buckets (hereafter: mesocosms) by individual hoses in an upwelling design. Each bucket also contained an airstone with continuously bubbling air. We procured 1 year old *C. virginica* from The Choptank Oyster Company (Cambridge, MD, USA) and collected *M. balthica, A. mitchelli*, and *I. recurvum* from the Rhode River. We manipulated the bivalve community, with either just *C. virginica* (monoculture) or *C. virginica* and the other three bivalves polyculture). We crossed this cohabitation treatment with a density treatment, placing either 30 (low) or 60 (high) oysters in the monoculture mesocosms and either 15 oysters, 15 mussels and three clams (low) or 30 oysters 30 mussels and five clams (high) in the polyculture mesocosms.

We had nine replicate mesocosms per treatment combination (2 densities × 2 diversities), for a total of 36 mesocosms (Fig. S3). Salinity of the raw Rhode river water ranged from 4–6 ppt over the course of the experiment. Clams were buried in sand at the bottom of the buckets, while oysters and mussels were held in baskets 5 inches off of the bottom allowing the feces and pseudofeces to fall below the organisms (Fig. S3). Baskets had ample room and did not restrict the growth of the bivalves. Buckets were checked daily and any clogged hoses were unclogged to ensure water was flowing continuously. Buckets were thoroughly cleaned weekly: feces and pseudofeces were removed, bucket sides were scrubbed, and bivalves were rinsed off.

In June 2018, 15 of each species for oysters and mussels and all clams per mesocosm were weighed and measured as in the 2017 experiment and tagged with an individual bee tag. Mesocosms and hoses were cleaned weekly throughout the summer to avoid clogging and buildup of waste until October 2018. At this point, all organisms were removed from the experiment and counted. Seven tagged oysters, five tagged mussels, and all surviving clams from three randomly chosen replicate mesocosms (per treatment combination) were weighed, measured, and sacrificed. Gill, mantle, and digestive gland were again sampled for microbiome characterization.

## DNA extraction, 16S amplification and sequencing

DNA was extracted using the Qiagen DNeasy blood and tissue kit in 96 well plate format following manufacturers protocols. DNA was diluted 1:100 with water to reduce the amount of template and increase PCR success. We first amplified the V4 region of 16S rRNA gene using modified versions of the primers used by the earth microbiome project: 515-F (*Parada, Needham & Fuhrman, 2016*) and 806-R (*Apprill et al., 2015*). The primers were modified using a 20 bp addition that acts as a priming site for a second indexing PCR. Four sets of these primers were used that have 0–3 'N' base pairs added onto the 3′ end before the adapter to increase library heterogeneity and sequencing success (Table S1; *Fadrosh et al., 2014*). The PCR reaction consisted of the following: 0.125 µl of Taq Gold (Thermo Fisher Scientific, Waltham, MA, United States), 2.5 µl buffer, 0.5 µl 10 mM dNTPs, 1.5 µl MgCl2, 1 µl 10 µM forward primer, 1 µl 10 µM reverse primer, 1 µl DNA, 17.125 µl water, 0.25 µl BSA. PCRs were run using the following program: 95 °C for 10 mins, 35 cycles of 95 °C for 15 s, 50 °C for 1 min, 72 °C for 1 min, followed by extension at 72 °C for 10 mins. All PCRs were performed in triplicate and pooled as described in *Lohan et al. (2016)*. We then used unique combinations of Nextera Illumina indexes in a second PCR reaction consisting of the following: 9.5 µl water, 12.5 µl KAPA ReadyMix, 1 µl of 10 µM forward and reverse indexing primers, 1 µl of pooled PCR product. PCRs were run using the following program: 95 °C for 5 min, 12 rounds of 98 °C for 20 s, 60 °C for 45 s, 72 °C for 45 s, followed by extension at 72 °C for 5 min. Each indexing reaction was cleaned using Ampure XP PCR cleanup beads in a 1.8:1 ratio of beads to PCR product. The concentrations were then measured using a Qubit (Thermo Fisher Scientific, Waltham, MA, United States) and products were pooled in equal amounts. The final library was bead cleaned again. Libraries were sequenced on a MiSeq using paired end 300 bp sequencing by

the Laboratories of Analytical Biology at the Smithsonian National Museum of Natural History.

## Bioinformatics and data analysis

The raw 16S sequences described here are accessible *via* SRA under the Bioproject PRJNA1047489. Primers were removed using cutadapt (*Martin, 2011*) and resulting reads were then run through DADA2 (version 1.24.0, *Callahan et al., 2016*) where they were filtered and trimmed using a maximum ee cutoff of 2, truncation quality cutoff of 2, and truncation length of 250 and 230 in the forward and reverse reads. Trimmed reads were then clustered into ASVs, merged using a minimum overlap of 20, maximum mismatch of 0, and chimeras were removed. Samples with fewer than 10,000 reads were excluded. Taxonomy was assigned using the Silva database, version 138 (*Quast et al., 2012*). ASVs that were identified as belonging to the kingdom Eukaryota or order Chloroplast were removed. We used the R package phyloseq (*McMurdie & Holmes, 2013*) version 1.36.0 to combine the microbial taxonomy, amplicon sequence variant (ASV) table, and metadata of each sample. ASV count data were normalized by dividing by total counts for each sample and multiplying by $1 \times 10^6$.

Because of the large differences in location, environment, and experimental design, between 2017 and 2018 samples, we only did one analysis with the combined samples, to look at community level differences: PERMANOVA of Bray-Curtis dissimilarities (function adonis in package Vegan version 2.6.2; *Oksanen, Guillaume Blanchet & Kindt, 2013*), and corresponding nMDS plot visualized using phyloseq. All subsequent analyses were performed on 2017 and 2018 samples separately.

We determined the significance of multiple factors (site, RFTM score, and peacrabs for 2017; density and species for 2018; cohabitation and density for 2018 oysters) on the overall microbial community using PERMANOVA and nMDS.

We identified ASVs associated with different factors using the R package DESeq2 version 1.32.0 (*Love, Huber & Anders, 2014*). We counted an ASV as significantly associated if it returned a Benjamini-Hochberg adjusted $p$ value of less than 0.05 and was present in more than one third of the samples (filtered using genefilter_sample( )).

To look for ASVs associated with oyster size in both 2017 and 2018 data, we calculated delta length and weight as follows: [(pre-post)/pre]. Significant ASVs were identified separately for each year as described above. For 2017 oysters, RFTM and pea crab abundances were converted into presence/absence data (0 and 1). We identified OTUs significantly associated with parasite presence using DESeq2 with a model of ~RFTM +Peacrab+Site and separating ASVs associated with *P. marinus* presence/absence and *Z. ostreum* presence/absence. Finally, we identified ASVs associated with cohabitation in 2018 oysters using DESeq2 as described above.

For the resulting significant ASV tables, the fill function from the tidyr package (version 1.2.0) was used in a 'for loop' to fill in the taxonomic ranks with the lowest rank identified for each factor. The log2FoldChange column from the DESeq2 results function was inserted into each of the filtered taxonomy tables and plotted using ggplot2 (version 3.5.1; *Wickham, 2009*). The taxonomic heat trees were generated by inserting the pruned
**Table 1 PERMANOVA results for both 2017 and 2018 data.** An asterisk (*) denotes a significant result.

| Factor | Df | SumOfSqs | R2 | F | Pr (>F) |
|---|---|---|---|---|---|
| Experiment | 1 | 7.36 | 0.12 | 29.72 | 0.001* |
| Experiment:species | 3 | 10.06 | 0.16 | 13.53 | 0.001* |
| Residual | 207 | 46.34 | 0.73 | NA | NA |
| Total | 211 | 63.77 | 1 | NA | NA |

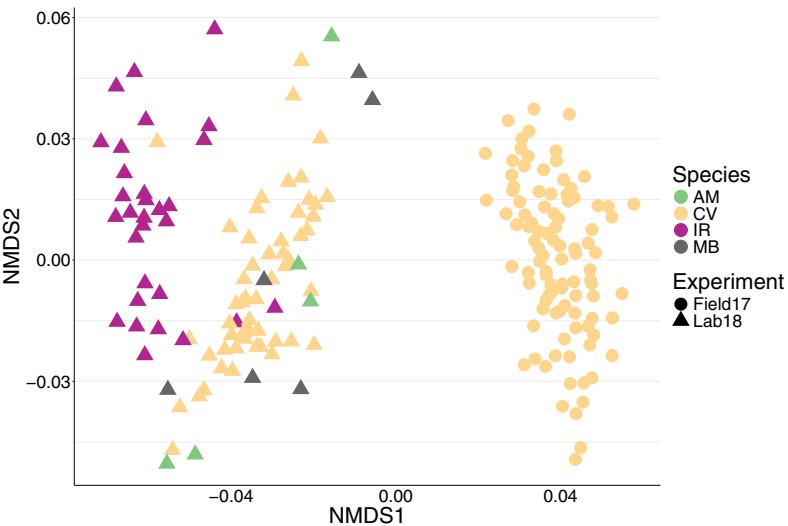

**Figure 1 nMDS of 192 bivalve microbiomes (combined mantle, gill, digestive gland) showing clustering by species and experiment.** Species are distinguished by color and experiment by shape. nMDS converged after 20 permutations, 2D stress = 0.1539981.

taxonomic table for each factor into the parse_phyloseq ( ) function and then plotted using the heat_tree ( ) function of the metacoder package (version 0.3.5). GitHub repository with code is available here: https://github.com/sagw/Oyster_16S/tree/main.

# RESULTS

## Differences between experiments

We found 12,999 ASVs across 192 bivalves from both (2017 and 2018) datasets. We saw a strong effect of experiment on the bivalve microbiome (Table 1, Fig. 1), with the nMDS showing that the oysters from the 2017 field experiment clustered separately from all samples from 2018 regardless of species (Fig. 1). There were several factors that likely lead to the differences in microbiomes between 2017 and 2018 oysters: 2017 oysters were purchased from a farm in Wachapreague, Virginia and placed *in situ* at sites around Wachapreague where salinity is consistently above 25 ppt (see methods), whereas the 2018 bivalves were either purchased from a farm in Harris creek, Maryland (oysters), or

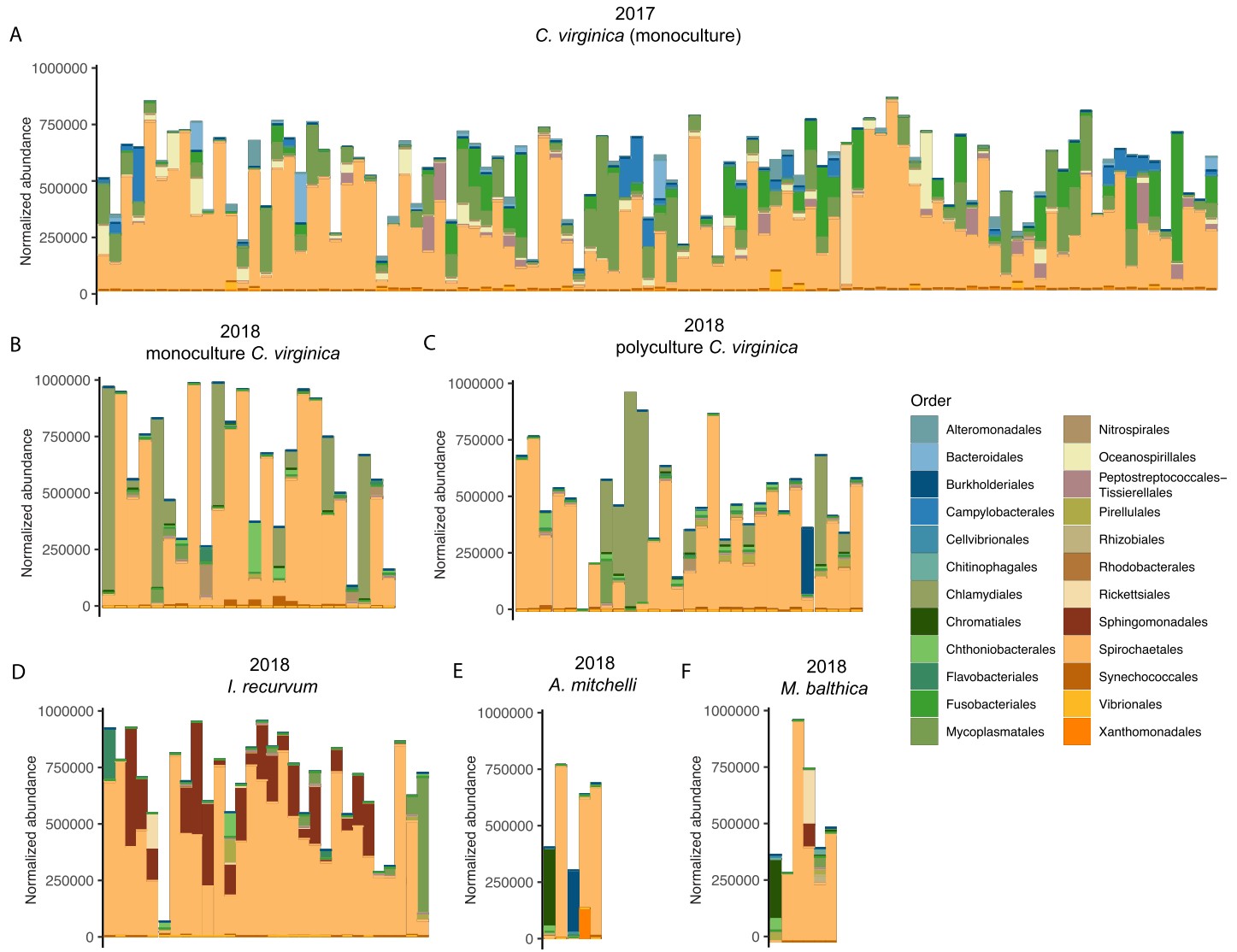

**Figure 2** **Top 10 most abundant orders found in the microbiomes (combined mantle, gill, digestive gland) for each group.** (A) 2017 *C. virginica*, (B) 2018 monoculture *C. virginica*, (C) 2018 Polyculture *C. virginica*, (D) 2018 *I. recurvum*, (E) 2018 *M. mitchelli*, (F) 2018 *M. balthica*. Y axis is normalized abundance (abundance divided by total abundance for that sample × 10⁶).

collected from the waters around SERC in Edgewater, Maryland (mussels and clams) and held in Rhode River water, which is below 10 ppt.

## Characterizing the microbiomes of *C. virginica*, *Ischadium recurvum*, *Macoma balthica*, and *Ameritella mitchelli*

We saw a strong effect of host species (nested in experiment) on the bivalve microbiome (Table 1, Fig. 1). Species accounted for slightly more of the variation than the experiment alone (Table 1). Within the 2018 lab experiment, clams generally clustered with oysters, while mussels clustered separately (Fig. S4). At an ASV level, we found clear differences between the microbiomes of the four species surveyed. However, looking at just the top 10 most abundant orders for each group (2017 oysters, 2018 oysters, mussels, and the two

clam species), all four species had strong similarities in their microbiome. We found no significant difference in absolute richness (total number of ASVs) across bivalve species (Fig. S5). Most individuals, regardless of species or experiment, were dominated by bacteria in the order Spirochaetales (Fig. 2). Several of the 2017 oysters had high relative abundances of bacteria in the orders Fusobacteriales and Mycoplasmatales. In addition, 2017 oysters in general had higher abundances of Oceanospirillales, Alteromonadales, and Campylobacterales than other groups. In contrast, several 2018 oysters were dominated by Chlamydiales, with three oyster microbiomes made up of >50% Chlamydiales. Most mussels had higher abundances of bacteria in the order Sphingomonadales, and several had higher abundances of Flavobacteriales than other groups. One *A. mitchelli* clam was dominated by Alteromonadales, and one was dominated by Campylobacterales (Fig. 2). It should be noted that these organisms were held in mesocosms and therefore these results may not be reflective of their microbiomes in the wild. The *I. recurvum* microbiome had significantly less dispersion than that of *C. virginica* individuals (2018 only), indicating that *I. recurvum* individuals were more similar to each other (Permutation test for homogeneity of multivariate dispersions, $p < 0.001$, Fig. S6).

All 151 *C. virginica* across both experiments shared a single core ASV, identified as belonging to the family Spirochetaceae (Table S2). This ASV was the only core microbe found across all 98 2017 oysters and 53 2018 oysters when analyzed separately. In addition, this ASV was shared by all eleven clams of both species. Twenty-nine out of the thirty mussels also contained the oyster core ASV, although for most at very low abundances (Fig. 3). One mussel's microbiome was dominated (97%) by the core oyster ASV. All 30 *I. recurvum* shared a single different ASV also identified as belonging to Spirochetaceae (Fig. 3). All 11 clams of both species shared a single ASV identified as belonging to the family *Pirellulacea* (Table S2, Fig. 3). The *A. mitchelli* also shared three other core ASVs, two identified as Cyanobium PCC-6307 and one in the genus Candidatus Megaira in the family Rickettsiaceae (Table S2).

## ASVs associated with bivalve change in size

Overall, oysters in 2017 increased more in length and weight than either 2018 oysters or mussels (Figs. 4A and 4B, Table S3); however, we did not identify any ASVs significantly associated with change in length or weight for 2017 oysters and did not find any ASVs associated with change in length across all samples. There were twelve ASVs significantly positively associated with change in weight for 2018 oysters (Figs. 4C and 4D). Three of these ASVs were in the class Actinobacteria and two were in the genus Cyanobium PCC-6307. In addition, we found three ASVs positively associated with 2018 mussel weight, two of which were in the family Pirellulaceae (Figs. 4E and 4F).

## Effect of site on the oyster microbiome

For oysters from the 2017 experiment, only site had a significant effect on the overall microbiome, explaining 7.5% of the variation (Table 2). The nMDS shows the three sites segregating from each other with site OY more different than the other two (Fig. S7).

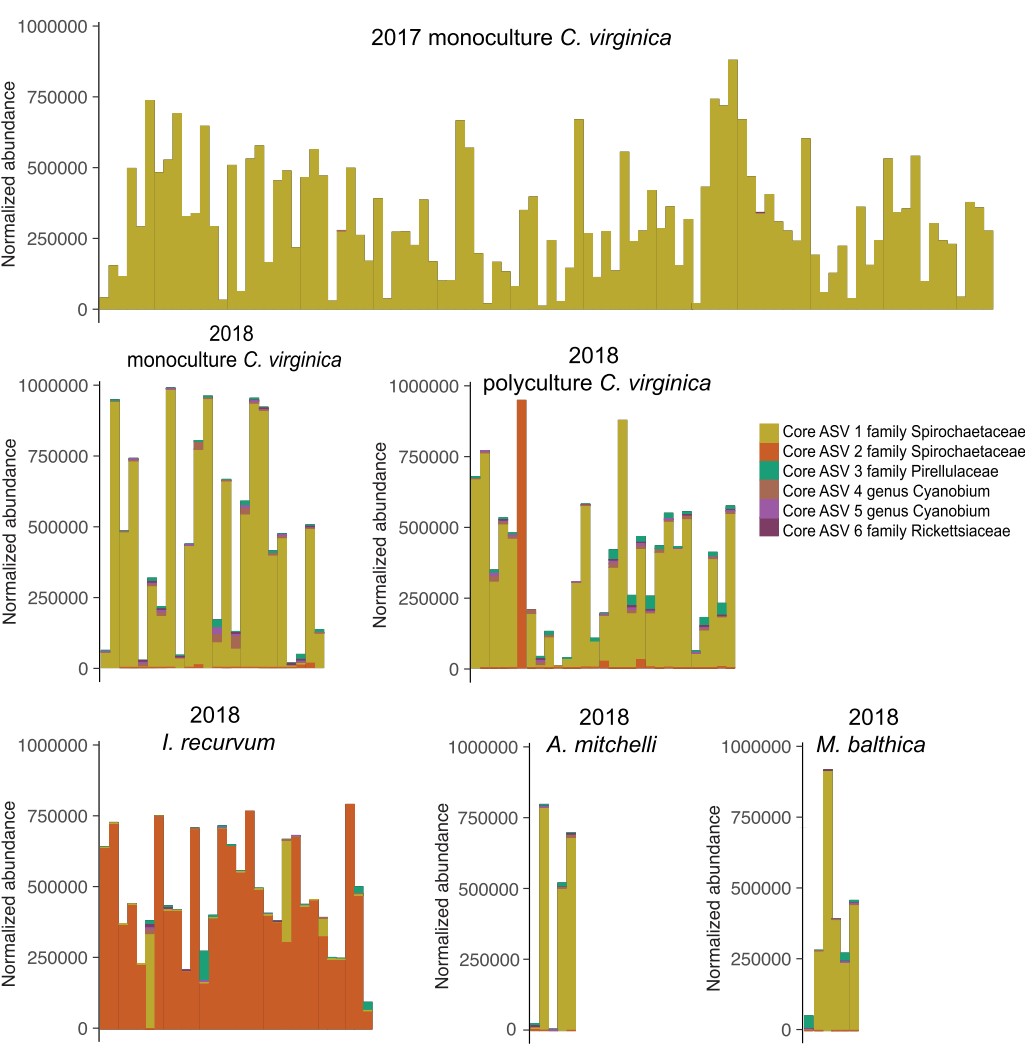

**Figure 3 Normalized abundance of core ASVs across species and year.** Y axis is normalized abundance (abundance divided by total abundance for that sample × $10^6$). Additional detail about each core ASV is in Table S1.

## Effect of density and cohabitation on the oyster microbiome

Neither density nor cohabitation had a significant effect on the overall oyster microbiome (Table S4), however one oyster in the polyculture treatment clustered with the mussels. One ASV, which was identical to the mussel core Spirochete (and also negatively associated with oyster growth) was positively associated with both cohabitation (more abundant in cocultured oysters, log2 fold change of 2.26) and density (log2 fold change of 2.14). Another ASV belonging to the family Mycoplasmataceae was negatively associated with density (log2 fold change of −4.82).

## Effect of parasitism on the oyster microbiome

Seventy percent of 2017 oysters ($n = 78$) were infected with *P. marinus*, while eight percent ($n = 9$) were infected with peacrabs (Table S3). Although at a community level, infection by *Perkinsus marinus* did not have a significant effect on 2017 oyster microbiomes, 11 ASVs

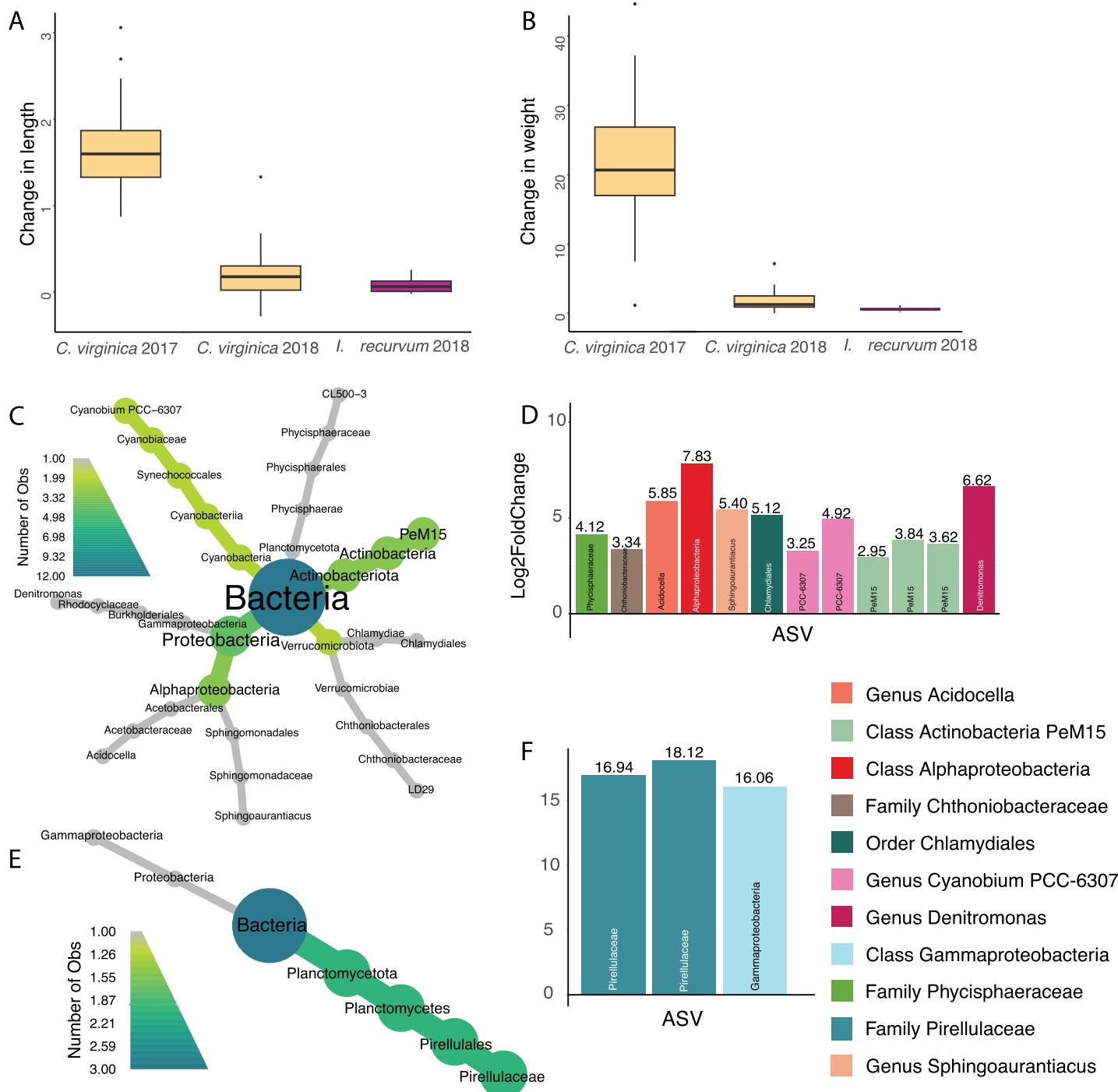

**Figure 4  Change in size for 2017 and 2018 bivalves and associated ASVs.** (A) Boxplot of change in weight (post-pre/pre) in 2017 and 2018 oysters and 2018 mussels. (B) Boxplot of change in length (post-pre/pre) in 2017 and 2018 oysters and 2018 mussels. (C) Heat tree showing full taxonomy and (D) Log2 fold change of ASVs significantly associated with change in weight in 2018 oysters (combined mantle, gill, digestive gland). (E) Heat tree showing full taxonomy and (F) Log2 fold change of ASVs significantly associated with change in weight in 2018 mussels (combined mantle, gill, digestive gland). A positive log fold change indicates the ASV is more abundant in larger individuals.

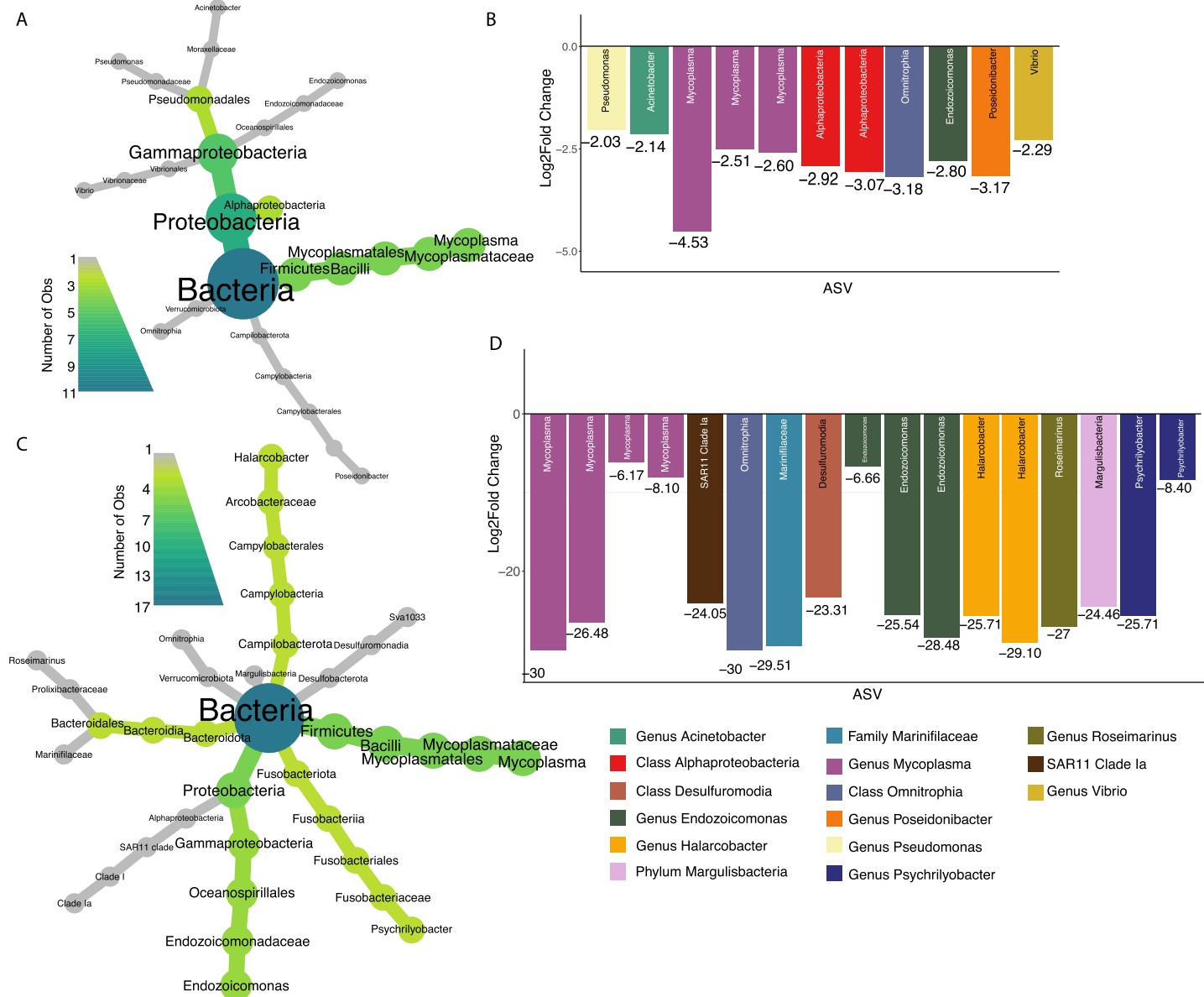

**Figure 5** **ASVs associated with parasitism by *Z. ostreum* and *P. marinus*.** (A) Heat tree showing full taxonomy and (B) log2 fold change of ASVs significantly associated with *P. marinus* infection (presence-absence) in 2017 oysters (combined mantle, gill, digestive gland). (C) Heat tree showing full taxonomy and (D) log2 fold change of ASVs significantly associated with *Z. ostreum* infection (presence-absence) in 2017 oysters (combined mantle, gill, digestive gland). A negative log fold change indicates the ASV is less abundant in infected individuals.

were found to be negatively associated with *P. marinus* parasitism, with log2 fold changes ranging from −2.03 to −4.53 (Figs. 5A and 5B). None of these significant ASVs overlapped with those associated with change in length or weight. Three ASVs were identified as belonging to the genus *Mycoplasma* of the phylum Firmicutes. One ASVs was identified as *Vibrio*, and one ASV belonged to each of the genera *Pseudomonas, Acinetobacter*, and *Endozoicomonas* all of the phylum Proteobacteria. In addition, one ASV was identified as *Poseidonibacter* of the phylum Campylobacterota.

Seventeen ASVs were significantly negatively associated with *Zaops ostreum* infection. Thirteen out of 17 ASVs had low log2 fold changes between −23.3 and −30. Five of these ASVs (three in the genus *Mycoplasma* one in the genus *Endozoicomonas*, and one in the class Omnitrophia) were also significantly associated with *P. marinus* infection (Figs. 5C and 5D).

## DISCUSSION

We sought to characterize the microbiomes of four Mid-Atlantic bivalves under multiple conditions. We found that several factors influence microbiome community composition including host species, location, community, size, and infection status. Our findings provide a starting point for future studies identifying clear links between location, bivalve health and the microbiome.

### Chesapeake bivalve microbiomes are shaped by their environment and host species

We found that the environment played a large role in shaping bivalve microbiomes. Oysters from the 2018 experiment clustered with mussels from the same year rather than with oysters from the 2017 experiment (Fig. 1), suggesting that the environmental conditions (*e.g.*, temperature, salinity) and biotic components of the seawater (*e.g.*, microbes, algae) play a large role in shaping the bivalve microbiome. We expect that the strong salinity differences between Wachapreague (high salinity, 2017 experiment) and Edgewater (low salinity, 2018 experiment) likely played a larger role than other environmental parameters in structuring the microbial communities. In addition, while bivalves in the 2017 experiment experienced field conditions, in the 2018 experiment organisms were kept in buckets which likely had an influence on the microbiome. Furthermore, the relatively low salinity that organisms in the 2018 experiment experienced is reflective of conditions in the Rhode River, in the northern section of the Chesapeake Bay. The low salinity conditions likely account for the much slower growth of these oysters, indicating relatively poor health. Previous work by *Pierce & Ward (2019)* in Long Island Sound found relatively high similarity in gut microbial communities between *C. virginica* and the mussel *Mytilus edulis*, credited to their shared feeding mechanisms and environment (*Pierce & Ward, 2019*). Furthering this point, we found a significant effect of site on the oyster microbial communities in 2017, which accounted for 7% of the variation in microbiomes (Table 2). We did not find that parasite infection had a significant effect on the overall microbial community. These findings are consistent with a prior study showing site, but not infection with *P. marinus*, to have a significant effect at the community level on *C. virginica* microbiomes (*Sakowski, 2015*). Overall, our observed dominant effect of experiment and site support previous studies showing a strong effect of the environment on bivalve microbiomes (*Unzueta-Martínez, Welch & Bowen, 2021*), however, while many sites differ in salinity, we do not know of any studies that test the effect of salinity on oyster microbiomes in a controlled laboratory setting.

All four species had strong similarities in their microbiomes at a high taxonomic level (Fig. 2), although each species' microbiome differed at finer taxonomic scales (ASV). Most
**Table 2  PERMANOVA showing the effect of location and infection on the oyster microbiome (2017 field experiment).**

| Factor | Df | SumOfSqs | R2 | F | Pr (>F) |
|---|---|---|---|---|---|
| Site | 2 | 1.76 | 0.075 | 3.86 | 0.001 |
| RFTM_score | 1 | 0.28 | 0.012 | 1.21 | 0.23 |
| peacrabs | 1 | 0.32 | 0.014 | 1.42 | 0.10 |
| Site:RFTM_score | 2 | 0.55 | 0.023 | 1.20 | 0.19 |
| Site:peacrabs | 2 | 0.19 | 0.0079 | 0.81 | 0.65 |
| RFTM_score:peacrabs | 1 | 0.15 | 0.0062 | 0.64 | 0.86 |
| Site:RFTM_score:peacrabs | 2 | 0.20 | 0.0083 | 0.86 | 0.58 |
| Residual | 88 | 20.08 | 0.85 | NA | NA |
| Total | 97 | 23.52 | 1 | NA | NA |

individuals, regardless of species or experiment, were dominated by bacteria in the order Spirochaetales, which have previously been associated with marine invertebrates including *C. virginica* (*Pierce & Ward, 2019*). *Pimentel et al. (2021)* also found *C. virginica* microbiomes contained high relative abundances of spirochetes, with the highest abundances in the mantle and second highest in the gill. Since our samples contained both mantle and gill tissue, we assume most of the identified spirochetes were in these tissues. Spirochetes have been found in a wide variety of bivalve species including the Asian clam, *Corbicula fluminea*, and the intertidal mussels *Perna perna*, and *Chroromytilus meridionalis* (*Zhao et al., 2022*; *Simon & McQuaid, 1999*). They have previously been associated with disease in the Akoya pearl oyster, *Pinctada fucata* (*Matsuyama et al., 2017*), but are more commonly identified as symbionts contributing to nitrogen fixation and nutrient recycling (*van de Water et al., 2016*). Several of the 2017 oysters had high relative abundances of bacteria in the order Fusobacteriales (Fig. 2), which *Pierce & Ward (2019)* found in both bivalve guts and marine aggregate microbiomes. In addition, several 2017 oysters had high relative abundances of bacteria in the order Mycoplasmatales (Fig. 2), which have been associated with the gut microbiomes of *C. virginica* (*King et al., 2012*; *Pierce & Ward, 2019*; *Pimentel et al., 2021*), as well as other species of bivalves (*Akter et al., 2023*; *Arfken et al., 2017*; *Kunselman et al., 2022*). *Pimentel et al. (2021)* found that *Mollicutes* were significantly enriched in *C. virginica* gut microbiomes compared to biodeposits, suggesting a persistent association and potential mutualism with the gut. Analysis of their metagenome assembled *Mycoplasma* genome identified an arginine deiminase pathway which may contribute to the production of ATP as well as a putative chitinase; however, whether the relationship of these bacteria with their host is mutualistic or pathogenic remains unknown (*Pimentel et al., 2021*).

To our knowledge, this is the first study of the microbial composition of *I. recurvum, A. mitchelli*, and *M. balthica*. The mussel microbiome had significantly less dispersion than that of the oysters or *M. mitchelli* (Fig. S6). Mussel microbiomes were generally dominated by Spirochetes and most mussels had higher abundances of bacteria in the order Sphingomonadales than were found in oysters across either year (Fig. 2), but since this is

the first description of the *I. recurvum* microbiome, we do not know if they are common community members across locations. Sphingomonadales have been associated with other bivalve microbiomes, such as the *Anadara broughtoni* clam and (*Romanenko et al., 2007*), *Villosa nebulosa* mussels (*Aceves et al., 2018*) and the Pacific oyster *Crassostrea gigas* (*Wegner et al., 2013*). Several mussels also had higher abundances of Flavobacteriales than other groups (Fig. 2). Flavobacteria are common beneficial and pathogenic associates of marine animals (*Gignoux-Wolfsohn & Vollmer, 2015*; *Pierce & Ward, 2019*; *Egan, 2022*). We saw high variability in the microbiomes of both clam species, with no particular trends (Fig. 2). One *A. mitchelli* was almost entirely dominated by *Altermonadales* while another was dominated by *Campylobacter*, other individuals of both species had microbiomes similar to *C. virginica*. Both species of clams are facultative deposit feeders, able to both filter nutrients from the water column and consume material in the sediment. This feeding method may contribute to both the similarity of their microbiomes to those of *C. virginica* as well as the individual variability. Larger sample sizes are needed to determine whether these species' microbiomes are actually highly variable across individuals, given that this is the first description of either species' microbiome.

## Core microbes differ between oysters and mussels

We found core ASVs in the family *Spirochaetaceae* for each species, with two clam species sharing a core ASV with the oysters and the mussels containing a different core *Spirochetaceae* strain (Table S2). The oyster core ASV is 100% identical to the *Spirochaetia-1* found by *Pimentel et al. (2021)* at a median relative abundance greater than 10% in oysters from an aquaculture farm in Rhode Island (*Pimentel et al., 2021*). This strain may therefore form a persistent and conserved relationship with *C. virginica*, but more research is needed to understand how widespread this strain is across the host range. Interestingly, in our study the core mussel ASV was also negatively associated with change in length for oysters, suggesting that oysters dominated by their species-specific core ASV may be more fit than those colonized by the core ASV of a sympatric species. For most individuals, the species-specific core ASV was the most abundant, further suggesting it is likely an important member of its host's microbiome.

While most individuals were dominated by their species' core ASV, two mussels and one oyster in the polyculture treatment had very high relative abundances of the other species' ASV. These individuals clustered with the other species on a nMDS (Figs. 1, S7), indicating that the strain of spirochete is the largest factor differentiating species' microbiomes. While we did not find a significant effect of cohabitation (monoculture *vs.* polyculture) on the overall microbiome, we did find that the mussel core ASV was positively associated with both density and cohabitation, suggesting that when oysters are in close proximity to mussels, the abundance of this ASV increases. None of the 2017 oysters contained the core ASV associated with *I. recurvum*, possibly because this species of mussel is not found in the area where the 2017 experiment took place. While multiple studies have found shared microbes across cocultured bivalve species (*Pierce & Ward, 2019*; *Akter et al., 2023*), these microbes may be shared due to similar diets and water being filtered (*Cranford, Ward & Shumway, 2011*). Here, we show evidence that supports

microbes moving between bivalve species that live sympatrically in the wild, although we were unable to test movement from oysters to mussels. Transmission of non-pathogenic and beneficial microbes between individuals of both the same (*Rose et al., 2023*) and different (*Mosites et al., 2017*; *Kates et al., 2020*) species has been demonstrated in other systems, predominantly in mobile, social animals. Better understanding of the extent to which microbes move between individuals on an oyster reef will have implications for how reef structure and community composition can influence host health and ecosystem function *via* the microbiome (*Le Roux, Wegner & Polz, 2016*).

## ASVs associated with change in weight

We found several ASVs positively associated with weight in 2018 oysters and mussels, but none in the 2017 oysters (Fig. 4). Our measurements of size (length and total wet weight) were used due to limitations in time and equipment, however these provide very rough metrics and we are therefore likely missing ASVs that play a role in oyster growth and metabolism. In addition, the wet weight method is prone to error due to differences in liquid held inside the bivalves and changes in weight due to spawning. While the oysters did appear reproductive at the end of the experiment, we do not know if they had spawned during the course of the summer. Future studies should use buoyant weight and dry weight of non-reproductive tissues for more accurate measurements of tissue and shell size.

Three ASVs significantly associated with increased weight belonged to the PeM15 lineage of Actinobacteria, a group commonly associated with eastern oyster stomachs and guts (*King et al., 2012*). One ASV was in the order Chlamydiales, which has also been identified as highly abundant in oyster guts (*Pimentel et al., 2021*). We found several cyanobacteria that associated with both oyster weight (*Cyanobium*) and mussel weight (Pirellulaceae) that are likely being eaten by the oysters and may contribute to increased size through added nutrients (*Weissberger & Glibert, 2021*; *Pierce & Ward, 2019*). Associations between gut microbiomes and weight could indicate differences in digestion or feeding leading to an increased overall body mass or increase in the mass of the gut. Future work weighing individual tissue types would help disentangle these associations. These ASVs positively associated with oyster size could be candidates for development of probiotics used to increase oyster size and health (*Yeh et al., 2020*; *Campa-Córdova et al., 2011*; *Jasmin et al., 2016*).

## Parasite infection alters oyster ASVs

All the ASVs significantly correlated with infection by either *P. marinus* or *Z. ostreum* had a negative relationship, meaning they were more abundant in uninfected individuals (Figs. 5A and 5B). This was contrary to our expectations, since in other marine systems, infection is often associated with an increased abundance of opportunistic pathogens (*Gignoux-Wolfsohn, Aronson & Vollmer, 2017*). However, given that both of these parasites have small, long-term effects on oyster fitness (*Smolowitz, 2013*; *Soudant, Chu & Volety, 2013*), the oyster may be able to protect itself against secondary infections. All the ASVs identified here are therefore associated with healthy parasite-free oysters, with five ASVs negatively associated with both parasites.

We found three ASVs belonging to the *Mycoplasma* genus negatively associated with both *P. marinus* and *Z. ostreum* infection (Figs. 5C and 5D). These results are similar to what has been reported previously, where *Mycoplasma* were much more abundant in the digestive gland and pallial fluid of oysters that were uninfected than in oysters infected with *P. marinus* (*Pimentel et al., 2021*). *Pimentel et al. (2021)* further suggest that *P. marinus* and *Mycoplasma* may compete for host-derived arginine producing the observed patterns of relative abundance. In other species, *Mycoplasma* show varying relationships to infection. *Green & Barnes (2010)* found that infection of *Saccostrea glomerata* with the protozoan parasite *Marteilia sydneyi* led to an increase in *Rickettsiales* and a decrease in abundance of other bacterial taxa including *Firmicutes* and *Mycoplasma*. An opposite association between *Mycoplasmataceae* and disease has been found in *Crassostrea gigas*, where it was linked to oysters that were more susceptible to Pacific Oyster Mortality Syndrome (POMS) in both hatcheries and field conditions, suggesting a role as an opportunistic pathogen (*Clerissi et al., 2020*). Overall, however, our results suggest that strains of *Mycoplasma* are associated with a lack of both parasites indicating that *Mycoplasma* may be a beneficial symbiont associated with increased fitness, however further study is needed to fully characterize this relationship.

One ASV in the genus *Endozoicomonas* (phylum *Proteobacteria)* was negatively associated with parasitism by both *P. marinus* and *Z. ostreum* (Fig. 5). Previous studies have found *Endozoicomonas* to be a ubiquitous member of aquatic microbiomes, present in a variety of organisms including corals, cnidarians, tunicates and fish (*Neave et al., 2016*). *Endozoicomonas* are believed to play a role in host nutrient recycling (*Neave et al., 2017*; *Tandon et al., 2022*), and disease resistance (*Gignoux-Wolfsohn, Aronson & Vollmer, 2017*). However, they have also been tied to pathogenicity in bivalve species including oysters, mussels, and clams across various regions globally (*Cano et al., 2018*, *2020*). *Sakowski (2015)* found an ASV in the Family Endozoicomonaceae to be negatively associated with *P. marinus* infected *C. virginica* extrapallial fluid (*Sakowski, 2015*). Another ASV belonging to the genus *Poseidonibacter* was negatively associated with infection by *P. marinus*. *Poseidonibacter* are marine bacteria in the family Campylobacteraceae that are associated with numerous marine animal hosts, but play an unknown role in this symbiosis (*Kim et al., 2021*).

An ASV identified as belonging to the *Pseudomonas* genus was found to be negatively associated with *P. marinus* presence. Species of *Pseudomonas* have been found to be dominant in the digestive tract and internal organs of marine mollusks (*Ortigosa, Esteve & Pujalte, 1989*). In addition, *Murchelano & Bishop (1969)* found *Pseudomonas* to be the most numerous genus among bacteria isolated from juvenile *C. virginica*. Other experiments have found certain *Pseudomonas* strains lead to decreased growth, morphological abnormalities, and even increased mortality in oyster larvae (*Brown, 1973*). In contrast, the *Pseudomonas* I-2, *P. synxantha* and *P. aeruginosa* species has been previously studied for their probiotic properties for multiple farmed species (*Chythanya, Karunasagar & Karunasagar, 2002*; *Van Hai, Fotedar & Buller, 2007*; *Aguilar-Macías et al., 2010*). Strains of *Pseudomonas* have been used as probiotics in aquaculture due to their ability to inhibit growth of *Vibrio* spp. (*Van Hai, Fotedar & Buller, 2007*; *Aguilar-Macías*

*et al., 2010*). An ASV in the order Pseudomonadales has been previously negatively associated with *P. marinus* infection (*Sakowski, 2015*). Further investigation of this ASV at the strain level is necessary to understand the role it may be playing in infected oysters.

We also found a negative correlation between a *Vibrio* strain and *P. marinus* infection (Figs. 5A and 5B). The *Vibrio* genus contains common zoonotic pathogens that have direct effects on human health if consumed in raw seafood. Previous studies have found contradicting relationships between *P. marinus* and several *Vibrio* species where a potential positive trend was present between *P. marinus* infection and *V. parahaemolyticus* but not for *V. vulnificus* or generalized *Vibrio* spp. indicating a possible strain specific effect (*Bienlien et al., 2022*; *Audemard et al., 2023*).

## CONCLUSIONS

We examined the bacterial microbiomes of four Mid-Atlantic bivalves across species, location, health status, and community composition, to identify members of the microbiome that are fixed (similar regardless of extrinsic and intrinsic factors) and those that are plastic (change in response to factors). This is the first study to describe the microbial composition of *I. recurvum, A. mitchelli*, and *M. balthica*. By comparing these unknown bivalve microbiomes to the more characterized *C. virginica* microbiome, we separated common associates of bivalves from species-specific strains. We found that most individuals of all four species were dominated by bacteria in the order Spirochaetales, frequently observed members of the *C. virginica* microbiome, which may be involved in nitrogen fixation. We further found core ASVs for each bivalve species that belonged to this family, with core ASVs shared between the two clam species and oysters, while the mussels contained a different strain. These core ASVs were the most abundant ASV for most of the individuals, suggesting they likely play an important role in their host's microbiome. Other differences between bivalve species and treatments include the high abundance of bacteria in the order Mycoplasmatales in 2017 oysters, which are common members of the oyster gut microbiome. Most mussels had higher abundances of bacteria in the order Sphingomonadales than were found in oysters across either year. In addition, individual mussels were more similar to each other than oysters were. We saw high variability in the microbiomes of both clam species, with no particular trends, warranting further study of these species. Similar to previous studies, we found a strong effect of the environment on bivalve microbiomes as evidenced by the community level effects of experiment and site. We also identified many individual ASVs whose abundances are affected by external and internal factors. We found that core mussel ASVs can increase in abundance in oysters when cocultured, supporting the idea that microbes can move between bivalve species on a reef. Similar to other studies, we found ASVs belonging to the genus *Mycoplasma* negatively associated with parasite infection, suggesting these bacteria may be beneficial symbionts associated with increased host fitness. We also found ASVs in the genus *Pseudomonas* negatively associated with *P. marinus* presence, supporting their potential value as oyster probiotics. More broadly, our finding that all ASVs significantly correlated with infection by either parasite had a negative relationship to parasitism supports the Anna Karenina hypothesis where the microbiomes of healthy individuals are

more similar than those of diseased (*Zaneveld, McMinds & Vega Thurber, 2017*). This article lays important groundwork for future studies on the microbes associated with these hosts and their role in host health and ecosystem function.

## ACKNOWLEDGEMENTS

We would like to thank P.G. Ross, Clinton Arriola, Alexandra Gignoux, and Sean Fate for help with fieldwork, Tim Rapine for supplying oysters and clams for the 2017 experiment, and Teresa Vaillancourt, Matilda Newcomb, Ruth DiMaria, Pat Santos Ciminera, Brenda Soler Figueroa, Kristen Larson, Ashley Arnwine, Jess Hardee, Alexandra Gignoux, Erik Holum, Kristina Borst for help with lab work.

### Funding

This work was supported by a Burch fellowship and Smithsonian Institutional Fellowship to Sarah Gignoux-Wolfsohn, Smithsonian Latino Initiatives fellowships to Monserrat Garcia Ruiz and Diana Portugal Barron, a Smithsonian Women's committee fellowship to Monserrat Garcia Ruiz and Hunterdon funds to Katrina Lohan and Gregory Ruiz. The funders had no role in study design, data collection and analysis, decision to publish, or preparation of the manuscript.

### Grant Disclosures

The following grant information was disclosed by the authors:
Burch Fellowship.
Smithsonian Institutional Fellowship.
Smithsonian Latino Initiatives Fellowships.
Smithsonian Women's Committee Fellowship.
Hunterdon Funds.

### Competing Interests

The authors declare that they have no competing interests.

### Author Contributions

- Sarah Gignoux-Wolfsohn conceived and designed the experiments, performed the experiments, analyzed the data, prepared figures and/or tables, authored or reviewed drafts of the article, and approved the final draft.
- Monserrat Garcia Ruiz analyzed the data, prepared figures and/or tables, authored or reviewed drafts of the article, and approved the final draft.
- Diana Portugal Barron analyzed the data, prepared figures and/or tables, authored or reviewed drafts of the article, and approved the final draft.
- Gregory Ruiz conceived and designed the experiments, authored or reviewed drafts of the article, and approved the final draft.
- Katrina Lohan conceived and designed the experiments, authored or reviewed drafts of the article, and approved the final draft.

## DNA Deposition

The following information was supplied regarding the deposition of DNA sequences:

The raw 16S sequences are available at SRA: PRJNA1047489.

## Data Availability

The data is available at GitHub and Zenodo:

- https://github.com/sagw/Oyster_16S/tree/revision

- Sarah Gignoux-Wolfsohn. (2024). sagw/Oyster_16S: Bivalve microbiomes are shaped by host species, size, parasite infection, and environment (revision). Zenodo. https://doi.org/10.5281/zenodo.11406212.

## Supplemental Information

Supplemental information for this article can be found online at http://dx.doi.org/10.7717/peerj.18082#supplemental-information.

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
