# Peer review of "Bivalve microbiomes are shaped by host species, size, parasite infection, and environment"

_PeerJ, doi:10.7717/peerj.18082_

## Round 0.1 · original submission · Major Revisions

Both reviewers find the subject matter relevant and some findings valuable but provide a range of comments on experimental design, data presentation, and content of introduction and discussion. Please respond to all the comments in a response letter and explain how the revised manuscript addresses each of the comments. In addition to the comments from the reviewers, please respond to and address the following comments in the revised manuscript:

L176-177 Describe more clearly what you mean by ‘four sets of primers that have 0-4 base pairs added onto the end’: Which end? Which bases? Four sets at the same molarity, or same concentration? Is there a reference to this method?
L192 State how many samples in total were included in the pooled library.
L196-197 What were the truncation positions in 3’ and 5’ end for forward and reverse reads?
L198 Which version of Silva did you use?
L198 How many sequence reads per sample (range, average) were left after denoising? Were any samples omitted?
L203-204 Please include versions of vegan and ggplot, and any other software packages where it wasn’t listed.
L217 ‘in a for loop’ – please correct any error in wording
L221 Please add a verb to the final sentence about Github link.
L254 Please clarify the method you used to identify a single core ASV for each of the three bivalve species. Please include any relevant output tables as a supplement and assure the analysis is included in the submitted code.

Figure 3 requires a legend for the colors.
Figure 4: F and G are not included as labels in the figure – please add.
Figure 5b: unclear what is meant by presence-absence, given the figure gives a scale from 1 to 4; thus the scale is not binary.

Fig.S2. What does the arrow on the right to the oyster bag stand for?
Fig.S3. Please increase the resolution of this figure.
Fig.S4. Please change the y-axis to ‘Richness’

·

Basic reporting

This manuscript reports the microbiomes of four bivalve species native to the East coast of the US, and identifies how the microbiome may be shaped by species and environment. Two experiments are reported: one field experiment with eastern oysters in which the authors seek to understand the impact of parasitism on microbiomes, and a cohabitation experiment with four species of bivalves.

The manuscript is in general well written, the justification for the research is sound, and the subject is relevant, since it provides some new data on factors influencing bivalve microbiomes. The figures, with some notable exceptions listed below are relevant and clear. I have some suggestions about the introduction and discussion that I consider would strengthen the paper.

Comments/Suggestions:
- Introduction and discussion: Would all these bivalve species cohabitate in the natural environment, or do they have separate/distinct ecological niches?
- Introduction Line 98. Pea crabs are pathogens of bivalves, not just oysters. Likewise, there are Perkinsus spp. that affect bivalves other than oysters.
- Clarify line 136 - what do you mean by “placed in a marked container within the oyster bag”? Would that container affect the growth?
- Several figure legends lack detail (Fig 1 and 2) or are missing (supplementary figures). More detail should be provided describing what is represented in each axis.
- I suggest that the raw data for infection and growth should be shown in the supplementary data.
- Figures 2 and 3 are hard to interpret/read. Since microbiome data is compositional, I suggest that the authors should represent these plots as percent of relative abundance. This would make reading the figure easier by being able to see some of the lower abundance orders.
- I would not use “diversity” when referring to the comparison between monoculture and polyculture; this was confusing to me (since diversity is also used for bacterial diversity). I suggest using instead “cohabitation”.
- SRA repository information is missing.

Experimental design

A strength in this manuscript is the combination of field survey data (for Eastern oysters) with a mesocosm cohabitation experiment for the four species. I have, however, some concerns about the analysis of the two experiments. These are as follows:
- The two experiments differ greatly in salinity (2017 - around 30 ps; 2018 - less than 10 psu). It is well known that the microbial (bacteria and other microbes) communities in seawater are greatly impacted by salinity, but that is barely discussed in the manuscript. For example, I am not sure why the authors mention evaluating the effect of community diversity on Perkinsus spp. prevalence in the 2018 experiment (line 150) since prevalence and intensity of Perkinsus species would be very low; it is well documented that these parasites do not proliferate at salinities below 15 psu. Moreover, salinity below 10 psu can also highly impact oyster growth and survival. Authors should consider these factors in the presentation of their results and the discussion.
- The mesocosm experiment comparing polyculture with monoculture only seems to include a control for oyster monoculture, but not the other species of bivalves. Therefore, conclusions on the effect of cohabitation with other species can only be done for oysters. Interestingly, I did not see a comparison of oysters in monoculture versus oysters in polyculture - did I miss it?
- Microbiome analysis: I am somewhat concerned about the low number of reads/ASVs in some of the samples included in the analysis, as well as the big spread in richness seen in figure S4. Did the authors look at the rarefaction curves for all samples to make sure they had enough good quality reads to warrant the inclusion of all these samples in the analysis?
Methods comments:
- The RFTM method detects all Perkinsus species, including Perkinsus marinus (host - oysters) as well as other Perkinsus species that also affect clams (e.g., Perkinsus chesapeaki). The authors should refer to Perkinsus spp. (plural) instead of Perkinsus sp. (singular) throughout the text.
- How was bivalve volume calculated? Please provide a reference for the method used.
- Please provided the versions of Phyloseq and Silva used in the analysis.
- Were there controls included in the microbiome analysis to ensure lack of contamination?
- Provide models used in the statistical analyses for alpha and beta diversity.
- PERMANOVA - What was the number of permutations used? What is the stress used in the visualizations of the NMDS?
- Most bivalve microbiome experiments remove classified as Eukaryota, mitochondria, and chloroplast from the analysis, and focus on bacterial ASVs that are more likely to be tissue associated bacteria as opposed to the algal food. Please justify the inclusion of these reads in the analysis.
- Related to this; why would chloroplast ASVs be negatively associated with growth? This is counterintuitive to me. Please discuss.
- Nearing et al. 2022 (https://doi.org/10.1038/s41467-022-28034-z) showed that DeSeq2 may not be the most appropriate method to identify microbial ASVs associated with particular conditions. Please justify the use of this method as opposed to other methods, such as LefSe or correlation analysis (e.g., spearman rank).

Validity of the findings

I found the comparison on the different bivalves in the mesocosm experiment interesting and novel (monoculture and polyculture); this I consider is the most novel aspect of this research.
- The results of these comparisons (poly versus mono), however, are not clearly shown and barely discussed, and I am not sure the experiments were properly replicated.
- I am also concerned that the low salinity in the 2018 experiment may impact the validity of the results. Based on the data shown for growth and microbial richness, it appears that the conditions in the mesocosm experiments were suboptimal, which would affect data interpretation and the relevance as it relates to field conditions (authors do briefly mention this in the discussion). It would have been useful to have some data on the microbiome of these species before the cohabitation experiment started (time 0, upon collection from the field) as well as from the field at the end of the experiment to compare and/or controls of mesocosms with mussels or clams alone. I do understand that this is not possible at this time, but I recommend caution in how to interpret the results from the experiment, considering its limitations. In particular:
- Provide some evidence on the health status of the animals.
- I would recommend caution when stating that “mussels may be less influenced by their local environment” (Line 364), since this effect may be due to differences in the health status of these different species (as described in lines 391 - 393), or competition in feeding (to mention a few other possibilities).
- I would also recommend caution when stating (line 410) that ASVs may move across species, unless a more formal comparison of the microbiomes of mussels between mono and polyculture is performed (but there is no control mesocosms with mussels or clams alone).

Other comments:
- Line 154 - I am extremely surprised at the fact that only one core ASV was shared between all 170 oyster samples. Is this due to the strong effect of salinity in the two experiments?
- Discussion: ASVs associated with growth - Why would you get a different result with weight than volume?

Reviewer 2 ·

Basic reporting

The paper is well written and the literature appears appropriate. Some descriptions of methods and interpretation of the results need revising. The data presentation needs major revision to make the data more easily understood to a broad audience. The sequence data is available online, but I do not see a reference to it in the text.

Experimental design

I have concerns about the design of the mesocosm experiment. The methods do not supply enough information to know if the microbiomes reported truly represent healthy animals living in containment at low salinity or if similarities are due to overcrowding and poor water exchange. The overarching research question of if shellfish living in the same area share microbiomes is not fully addressed with the experimental design. Further details are listed in my additional comments section.

Validity of the findings

Replication is sufficient and statistical analysis is appropriate where applied. Sequence data is available online (though a statement needs to be added to the text). I would like to see data for the weights/volumes and any size measurements taken as well as information on the P. marinus load from histology. This information would be a valuable supplement for those interested.
My concerns over the validity of the findings and overall conclusions arise from the overall study design in 2018 and lack of controls. I do not think these findings should be reported as natural microbiomes of clams and mussels due to questionable husbandry. If the authors can show evidence that the bucket conditions did not restrict flow and growth of the animals, the results may be valid.

Additional comments

Major comments:
1) I am unsure of parts of the methods as they are currently written. My primary concern is the mesocosm experiment and if the animals had sufficient water flow and access to food. Based on the limited information on volume and weight change over the experiment, it looks like the animals were not feeding well. If the 5 gallon buckets with 30+ animals in them did not have good flushing, this would not only stress the animals but also could increase bottle effects affecting the microbial community present and shared between shellfish species. A lack of flushing would logically move the microbiomes of animals in close confinement closer together, which could be why the microbiomes of the oysters and mussels were so similar. Without a baseline of what the microbiomes looked like at the start of the mesocosm experiment, it is difficult to understand how the microbiomes shifted. Poor husbandry/overcrowding would also explain the clam mortality.

I also question the sampling method for the tissues sampled. It seems the microbiome samples are a combination of gill, mantle, and digestive gland. If the gut (running through the digestive gland) was sampled some of the time but not always, this would also lead to differences in observed microbiome.

2) The reasoning behind the mesocosm experiment is stated in the introduction (mussels and oysters can share reef space; clams are often found in sandy areas near oyster reefs), but the execution of this as an experiment on microbiome sharing falls short. There don’t seem to be any T0 samples to know what the microbiome of the animals was before going into the buckets, so the reported data cannot actually be presented as a true microbiome of the mussels or clams. My assumption for why there weren’t differences in the microbiomes in co- vs monoculture goes back to a general lack of flushing in the buckets. A much more straight-forward study would be to just sample both oysters and mussels living together in a reef and then perhaps each growing independently on different farms.

3) The data presentation needs significant work. Figures 1-3 are lacking captions and several axes are missing labels or have unclear labels. Figures 4 and 5 seem overly complex for the data they are showing, and I am unsure of why they are based on weight and volume.

4) Considerable time is spent discussing ASVs related to oyster wet weight as a proxy for growth. I do not understand how assumptions can be made between 2 years with different populations of oysters in vastly different environments (open water high salinity and buckets with very low salinity) that the differences in microbiome members is responsible for growth differences.

Minor comments:
Line 91: Suggestion to add “Dermo” in parentheses the first time P. marinus is mentioned, as a quick reference for readers.
Line 131 and 154: For the purpose of reporting and future work, do you know what family lines of oysters were used? It is not necessary, but some may be interested if you have this information.
Lines 139-140: Were all 3 tissues combined into one sample? My original assumption was these were going to be tested separately, but I don’t see analysis by tissue type. I further assumed dissection of a small piece of each tissue, but now am wondering if it was a whole homogenized animal? Was the digestive gland taken to avoid the gut, or could gut have also been included?
Line 153: Do you have flow rates for the buckets? Did you supplement any food?
Line 161: I would expect the low salinity here to result in vastly different microbiomes from the 2017 field study. Were any samples taken at the time of collection/mesocosm initiation to see how the communities changed in these low salinity buckets?
Line 161 & 168: You said there were 9 replicate buckets per treatment, but then you say only three replicate buckets are measured. What happened to the other 24 buckets?
Line 197: I would write out words fully: Maximum, minimum, truncation
Line 209: I am interested if infection load has an effect as well, rather than presence absence. This may have been the easiest way to deal with the data, but curious if you have looked?
Line 221: A Github repository with code IS available here
Line 264: Volume and weight seems an odd proxy for growth here as the 2017 oysters were presumably placed before spawning and would have spawned out (and lost volume and weight) over the season. The lack of growth in 2018 makes me question the ability of the animals to feed in the mesocosm experiment.
Line 308: Just a note, it would be interesting if pursing this research further to look at the microbiome of the peacrabs to see if the significant ASVs are arising from the crab or in response to the presence of the crabs!
Line 331: I think this is a really interesting finding that could be strengthened if you dig deeper into the P. marinus data. It would be interesting to see if the animals that were highly infected (not just present, but had the highest loads) had any differences from the non-infected animals. If there is a gradient affect, it could dilute your stats by looking at presence/absence. If there really is no effect, perhaps a good way of showing it would be to color the infected samples on your nMDS in figure 1.
Lines 361-381: A caveat needs to be stated in this paragraph that this microbiome was based on the end communities present in a mesocosm experiment, not on samples found in the environment. Additionally, for the clams, only a few survived the experiment, so the data likely does not represent healthy animals.
Lines 417-421: I’m still not sure how you associated these ASVs with growth. Are they just different between 2017 and 2018 or are they specifically from larger oysters in 2017 over smaller oysters in 2017? There are so many differences between 2017 and 2018, that I don’t see how you can associate ASVs specifically to growth.
Lines 421-424: Yes. But in 2017 you mentioned the animals were measured as well at the start. Length should be a better proxy for growth than wet weight. Since these studies happened during spawning season, wet weight would be expected to drop anyway in healthy animals.
Include accession number for sequence data – I see it on the review site, but not in the manuscript.
Figures:
Fig. 1: Are all 3 tissue types included in these data? I may be missing it, but I don’t see a caption for Figs 1-3, just a title.
Fig. 2: The colors are impossible to distinguish here with the current spectrum used and the small size of the bars. I’m not sure that doing the 10 most abundant orders makes sense here, as it makes it look like something that may be included in one graph is completely missing for a different species. I would recommend showing all of the data and grouping lower count taxa into an “other” category. Also, I would reorganize and make this figure larger to really see the data. Perhaps 3 lines, to move the 2018 oyster graphs next to eachother.
Fig. 3: Again, I may be missing it, but I don’t see a caption to explain the figure. The title says relative abundance, but the axes are unlabeled and go above 1 or 100%. What are the colors? The light blue is also likely to be missed, I would recommend a darker color for that.
Fig. 4: B & D – I am not familiar with this type of “heat-tree,” but it seems like it is only showing the full taxonomy of a few ASVs. I don’t think these are adding much value for the space they take up. Overall, this figure seems difficult to interpret and seems like it could be made into a table that includes the log2 fold change values. I am also not sure if the ASVs chosen here are significant ASVs found in the higher volume animals in each year compared to lower volume animals or compared between years.
Fig. 5: I have similar concerns as in figure 4. I do want to see this data for dermo presented in some way, but I don’t think this is the most intuitive method. I’m also not sure why 5A is based on weight and the rest of the figure is based on presence/absence of dermo.
Fig. 6: The caption for 5B is the same as 6A yet the heat trees are showing different results. This figure serves more of a purpose than the two previous, as the heat trees actually have branching and more than 1 or 2 observations. I am still unsure based on this if the negative fold change is based on presence or absence. i.e. are these ASVs reduced when infected or increased with infection? This could use clarity in the caption.
Fig. S3 is low quality. This should be larger to read without zooming.
Fig. S4: I find it interesting that there are multiple oyster samples with low alpha diversity (seemingly on the 0 line). Did these sample affect the data/were they retained in analysis?
Both tables can be summarized in the text and include unnecessary information for a reader.

---

## Round 0.2 · Minor Revisions

Reviewer 1 has provided additional suggestions for your consideration for improvement. Please respond to the reviewer comments and respond to all of my additional comments below in a response letter.

1. You stated that some samples were omitted due to low read abundance. I recommend including a supplementary table with read counts (raw and filtered) for the 192 sequenced samples along with information on which samples were omitted from the set. The information could be added to current Table S2.

2. Reviewer points 1.22 and 2.2.
Comment: Please include a statement in the manuscript discussion noting the point about low salinity during experiments and relevance to the shellfish health.

3. Reviewer points 2.7., 2.13, 2.14
Comment: Please include a statement in the discussion considering the potential impact of incubation conditions in shifting the microbiome away from natural, and potential influence on study results.

5. Reviewer point 2.31
Comment: Please add to the caption of Figure 5 indicating what negative vs. positive log fold change stands for.

6. Figure 4
The figure has ‘E’ twice.

7. Figures 4-5
The text on top of the bars is redundant since there is a legend in these figures but also unreadable from a printed page due to small font size and dark background on some of the bars. Please remove the text from the bars (Fig 4D, 4E, 5B, 5D).

8. Add captions to all supplementary figures.

9. Line by line comments:
L95 ‘in buried in’: delete the first ‘in’
L105 correct sentence structure in: ‘One of the most widespread group of protozoan parasites are...’
L133 Indicate city, state and country for Cherrystone Aquafarms
L138 Please clarify: ‘placed in a marked large mesh 25 in2 container’. Please use SI units and indicate mesh size if you have it.
L154 please indicate Edgewater, Maryland, in the Fig S1.
L157 Please indicate city, state, and country for Choptank Oyster Company.
L179 How many of the 9 mesocosms per treatment were processed for DNA sequencing? Any specific reasoning how the sequenced mesocosms were chosen of all?
L186 Thermofisher> Thermo Fisher
L187-188 Use symbol for ‘m’ to indicate micro instead of writing ‘u’.
L197 Indicate company for ‘Qubit’.
L199 Miseq >MiSeq
L205 dada2 > DADA2
L229 presence absence data > presence/absence data
L235 delete ‘function’
L249 Wachapreague, VA > Wachapreague, Virginia
L251 Harris creek MD > Harris Creek, Maryland
L252 Rhode river > Rhode River
L276 C. virginica > C. virginica individuals
L365 and 367-368 high abundances of > high relative abundances of
L542-543 Add a citation to ‘Anna Karenina hypothesis’.
Figure S1. Increase resolution of this figure, and include Edgewater, MD.

10. Following comments are from the previous editorial decision. These comments, with line numbers referring to the original manuscript draft, also need to be addressed:

L176-177 Describe more clearly what you mean by ‘four sets of primers that have 0-4 base pairs added onto the end’: Which end? Which bases? Four sets at the same molarity, or same concentration? Is there a reference to this method?
L196-197 What were the truncation positions in 3’ and 5’ end for forward and reverse reads?
L203-204 Please include versions of vegan and ggplot, and any other software packages where it wasn’t listed.
L217 ‘in a for loop’ – please correct any error in wording

·

Basic reporting

The investigators have addressed all my comments and concerns; I only have some minor suggestions.
- Mention more clearly somewhere in the discussion (as a reminder) that the mesocosm cohabitation experiment condition replicate the environmental conditions of the Rhode River, where these bivalves can be found.
- Lines 464 - 469 - I suggest adding at least one reference (maybe a review on bivalve feeding) showing the similarities and differences in feeding strategies of different species of mussels, clams, and oysters. The Ward group at UConn has several recent studies that are relevant (e.g., Pierce & Ward, Griffin et al. several papers) that should provide a good reference within their papers.
- Line 480 - Maybe add a reference also supporting this? I believe LeRoux et al. 2016 (doi: 10.1016/j.tim.2016.03.006) would be relevant, but they do have others that could be useful (they focused on vibrio, but it could apply to other pathogens).
- Line 525 - Another possibility to think about - that these cyanobacterial ASVs are associated (and could potentially be used as markers) with a “good quality food” phytoplankton community. No need to add it, just a thought…
- Line 634 - Maybe add a final sentence saying that a more throughout characterization of these ASV at the species and strain level would be necessary. This is true for all ASV (as commented in the next paragroph for Vibrio, so you could make the statement in line 641 be more general for all ASVs associated with growth.

Experimental design

No comment

Validity of the findings

No comment

---

## Round 0.3 · Minor Revisions

Unfortunately, there are a few more items to fix before the manuscript can be further considered. The issues relate earlier comments on methods and data reporting. They will be important to address and clarify to the readers.

Earlier comment: You stated that some samples were omitted due to low read abundance. I recommend including a supplementary table with read counts (raw and filtered) for the 192 sequenced samples along with information on which samples were omitted from the set. The information could be added to current Table S2.

Response: Added to Table S2

New Comment: There are only 112 samples listed in Table S2 although 192 were reported as having been included in the dataset (above the 10,000 read cutoff) (Line 251). Please clarify the column content for this table by revising the column names and/or adding a clear caption on top row of the csv file. If the data are from field experiment of 2017, as it appears, what do the treatment density indicators ‘POLY’ vs. ‘MONO’ indicate for these field samples? This is confusing. It’s also unclear what the differences are between oysters within each site. The text discussed ‘10 oysters per bag’ – but bags are not separately indicated here. How many bags were there per site? Should different bags within site be considered/indicated differently?
It appears the sequence data for the mesocosm experiment including sequence yield for CV, AM, IR, and MB from 2018 are not included in this table. Please include all samples in this table including whatever data are applicable/available.

Figure 4
Earlier comment: The figure has ‘E’ twice.
Response: Corrected, thank you for noticing.
New Comment: The figure still has ‘E’ twice.

Figures 4-5
Earlier comment: The text on top of the bars is redundant since there is a legend in these figures but also unreadable from a printed page due to small font size and dark background on some of the bars. Please remove the text from the bars (Fig 4D, 4E, 5B, 5D).
Response: I added the text based on the reviewers feedback that the legend was not color blind friendly. While I agree it is redundant and not helpful when printed, if you don’t mind I would like to leave it so that if someone is colorblind they could blow it up on the computer and read it.
New Comment: However, in many cases the black text is unreadable on top of the dark bar color. White text would work better for most bars. Please revise so that any text on top of the bars is readable.

Earlier comment: L197 Indicate company for ‘Qubit’.
Response: Added
New comment: Was not added

Earlier comment: L276 C. virginica > C. virginica individuals
Response: Added
New comment: add space after ‘individuals’

Earlier comment: Figure S1. Increase resolution of this figure, and include Edgewater, MD.
Response: Fixed
New comment: Edgewater, MD is not shown in the revised map.

Earlier comment: L176-177 Describe more clearly what you mean by ‘four sets of primers that have 0-4 base pairs added onto the end’: Which end? Which bases? Four sets at the same molarity, or same concentration? Is there a reference to this method?

Response: Added clarifying language and reference line 189.

New comment: 0-4 sets of Ns sounds like there should be five, not four sets. The original paper used 0 to 7 bp spacer, which is still different, so reader will not know what primers were used by looking up the cited paper. Please include a supplementary table listing the primer sequences you used in this study.

---

## Round 0.4 · Minor Revisions

The authors need to address three remaining issues before the manuscript can be further considered.

1)
Comment on previous review (R2): There are only 112 samples listed in Table S2 although 192 were reported as having been included in the dataset (above the 10,000 read cutoff) (Line 251). Please clarify the column content for this table by revising the column names and/or adding a clear caption on top row of the csv file. If the data are from field experiment of 2017, as it appears, what do the treatment density indicators ‘POLY’ vs. ‘MONO’ indicate for these field samples? This is confusing. It’s also unclear what the differences are between oysters within each site. The text discussed ‘10 oysters per bag’ – but bags are not separately indicated here. How many bags were there per site? Should different bags within site be considered/indicated differently?
It appears the sequence data for the mesocosm experiment including sequence yield for CV, AM, IR, and MB from 2018 are not included in this table. Please include all samples in this table including whatever data are applicable/available.

Response, R2: Sorry I had a merging issue I didn’t notice. All the data are there now! I deleted confusing treatment names and clarified the column names for replicate bags and buckets.

New comment, R3: As stated in my earlier comment, it is confusing what the UniqueID names ‘high’ or ‘low’ ‘MONO’ and ‘POLY’ are referring to for the samples collected from the 2017 field experiment (column B in Table S3). It remains unclear what the differences were with respect to treatment type between the bags from 2017, if any. It appears bags were not separated in any way in the analysis however – only site differences were considered. Please clarify in the Methods section and the Table S3.

2)
Comment on R1: Figure S1. Increase resolution of this figure, and include Edgewater, MD.
Response, R1: Fixed

Comment on R2: Edgewater, MD is not shown in the revised map.
Response, R2: Yes it is, its not near wachapreague which might be the confusion? I made the E and W red to hopefully make them more visible.

New comment, R3: The reader doesn’t know what E and W stand for. Please add the information to the figure caption.

3)
New comment, R3:
Fix Supplementary Table legend: The caption for Table S4 currently lists it as Table S3.

---

## Round 0.5 · accepted · Accept

In the final submission, please add the caption to Figure S1 explaining W and E.